# Hypoxia in A Patient with Anti-p200 Pemphigoid under Combined Dapsone and Pantoprazole Treatment

**DOI:** 10.3390/biomedicines10112837

**Published:** 2022-11-07

**Authors:** Sebastian Lang, Philipp Wilhelm Sänger, Sandra Kocina, Christian von Loeffelholz

**Affiliations:** 1Department of Anaesthesiology and Intensive Care, Jena University Hospital, Friedrich-Schiller-University, 07747 Jena, Germany; 2Department of Pediatrics, Erlangen University Hospital, Friedrich-Alexander-University, 91054 Erlangen, Germany

**Keywords:** hemolysis, sepsis, NAFLD, G6PD-deficiency, CYP2C9, CYP2C19, proton pump inhibitors

## Abstract

A 70-year-old male patient was admitted to our dermatology outpatient clinic with newly developed personality changes and signs of hypoxemia. His anti-p200 Pemphigoid was treated with Dapsone for a few weeks. Due to generalized tonic-clonic seizure with a subsequent Glasgow Coma Scale of 5 points and a peripheral oxygen saturation not exceeding 88% under conditions of high-flow nasal cannula, he was intubated by the emergency team and transferred to the intensive care unit. Comprehensive tests were performed, but Dapsone-induced methemoglobinemia remained the exclusive explanation for the observed scenario, although arterial MetHb analysis showed a peak value of only 6%. The patient recovered shortly after repeated infusions of Methylene blue and Ascorbate, and cessation of Dapsone. We provide an overview of the pathophysiology, diagnostic procedures, and possible explanations for this case of Dapsone-induced methaemoglobinaemia. In conclusion, our case report provides evidence that even mild chronic methemglobinemia can induce severe clinical symptoms.

## 1. Introduction

Anti-p200 Pemphigoid is an autoimmune subepidermal blistering disease characterized by circulating and tissue-bound autoantibodies to a 200 kDa protein (p200) of the dermal–epidermal junction, considered to be important for the adhesion of basal keratinocytes to the underlying dermis. From a clinical perspective, patients typically present with tense blisters, urticarial papules and plaques, closely resembling bullous pemphigoid [1]. The effector mechanisms of anti-p200 Pemphigoid may involve the recruitment of pro-inflammatory cells such as lymphocytes and macrophages, and these may generate cytokines to promote the blistering or infiltration of neutrophils [2].

Dapsone (4,40-diaminodiphenylsulfone) is an aniline derivative with antibiotic and specific anti-inflammatory properties. This synthetic sulfone decreases folate synthesis by inhibiting the enzyme dihydropteroate synthetase. Dapsone is commonly used in the prevention of Pneumocystis jiroveci pneumonitis as an alternative agent in patients intolerant of or for whom trimethoprim/sulfamethoxazole is contraindicated [3]. Furthermore, Dapsone is an important drug in clinical dermatology to treat chronic skin diseases characterized by accumulation of neutrophils and/or eosinophils [4]. After oral ingestion, Dapsone undergoes extensive acetylation and hydroxylation in the liver, polmorphonuclear leukocytes, and monocytes [5]. After metabolization, it is subject to enterohepatic circulation and renal excretion. Dapsone metabolites are mainly hydroxylamines, resulting from specific hepatic transformation, and are considered a cause of major adverse effects, such as agranulocytosis, hemolysis, and methemoglobin (MetHb) formation. Importantly, glucose-6-phosphate dehydrogenase (G6PD)-deficient patients are reported to be more sensitive to hemolytic events, although less susceptible to MetHb formation [4].

## 2. Case Presentation

A 70-year-old Caucasian male with previously treated anti-p200 positive pemphigoid was brought to our dermatology outpatient clinic by his wife. Over the last four weeks, she had noticed personality changes in her husband, as well as an altered mental status and increasingly bluish lips and fingers. All these symptoms had become more pronounced in the last few days. About four weeks ago, an immuno-modulatory treatment with Dapsone had been initiated. Due to the severity of his symptoms, the patient was admitted.

No specific treatment was initiated at the inpatient dermatology, since shortly after admission he developed severe tachydyspnea and a generalized tonic-clonic seizure, followed by a prolonged postictal state with impaired consciousness (Glasgow Coma Scale [GCS] of 5 points). An episode of supraventricular tachycardia with a heart rate of 100 beats per minute was observed, while blood pressure remained within normal limits. On high-flow nasal cannula with 15 liters per minute, his peripheral oxygen saturation did not exceed 88%. Endotracheal intubation was performed by the hospital emergency team without complications. Head and chest CT scans were obtained (see Figure 1 and Figure 2).

The patient was then transferred to the interdisciplinary intensive care unit (ICU) of our university hospital. A comprehensive metabolic panel (CMP) was obtained, and an extensive workup for potential bacterial, viral and fungal infections was conducted.

In light of a recently initiated immune-modulatory therapy in a 70-year-old man, we were particularly concerned about complications from infectious diseases causing his symptoms. Therefore, we specifically evaluated him for community-acquired pneumonia and bacterial sepsis. The differential diagnosis included cardiac events such as myocardial infarction, various types of arrhythmia, pulmonary embolism, and cerebrovascular diseases. A pneumothorax was excluded (see Figure 2).

The patient’s past medical history was significant for arterial hypertension, carotid atherosclerosis, dyslipidemia, overweight (BMI 27.72 kg/m^2^), fatty liver disease (probably non-alcoholic fatty liver disease, NAFLD), gastroesophageal reflux disease, osteoarthritis of both knees, chronic back pain, and the above-mentioned anti-p200 Pemphigoid. His daily medications were Candesartan, Amlodipine, Pantoprazole, Dimentidene, Folate, and Dapsone.

There was no history of congestive heart failure, arrhythmias or coronary artery disease. The patient’s wife denied recent episodes of chest pain, palpitations, fatigue or edema. All chest imaging studies came back normal, without any signs of pneumonia or other significant pulmonary diseases (see Figure 2). The head CT scan showed no sign of an acute cerebrovascular event (see Figure 1). The standard ECG and CMP could not explain his symptoms. C-reactive protein (CRP) (8.9 (< 5) mg/L) and white blood cell count (WBC) (11.6 (4.4–11.3) gpt/L) were slightly increased on admission to the ICU, while procalcitonin (0.08 (< 0.5) ng/L) remained unremarkable. Urinalysis revealed leukocyturia, consistent with the mildly elevated CRP and WBC. PCR testing for SARS-CoV-2 was negative. No fever was detectable, and the differential white blood count analysis was within normal limits. Mild anemia was observed (hemoglobin (Hb) 6.9 (8.7–10.9) mmol/L).

Our tentative diagnosis of exclusion was Dapsone-induced MetHb-emia. On admission to the ICU, the patient’s arterial MetHb analysis showed a value of 6 (0.2–1.0)% (see Table 1). G6PD activity was inapparent (11 (7.0–20.5) U/g of Hb). There was no significant hemolysis, indicated by a nearly normal lactate-dehydrogenase (LDH) (4.2 (< 4.2) mol/L*s), and haptoglobine (0.3 (0.3–2.0) g/L).

In a setting of extensive controlled invasive mechanical ventilation (FiO2 1.0, Pmax 24 mB, PEEP 9 mB), arterial blood gas analysis was normal (see Table 1). In the absence of other organ dysfunction and due to the lack of alternative explanations, we decided to administer Methylene blue 2 mg/kg i.v. after consulting a toxicologist. A supportive therapy with Ascorbate 50 mg/kg/d i.v. was initiated. After 10 h, the arterial MetHb concentrations were decreased to 2.3%. We were capable of gradually reducing the intensity of mechanical ventilation and the patient was extubated on the subsequent day. His MetHb levels took several days to return to baseline (see Table 1) and required several doses of Methylene blue and Ascorbate i.v. each. Hb did not fluctuate significantly over the course of a few days of treatment, which was in accordance with our observation regarding hemolysis.

Twenty-four hours after extubation, the patient was able to walk independently, yet he showed some signs of mild delirium. He was evaluated by a neurologist and expected to make a full recovery. The patient also underwent ophthalmologic evaluation, which ruled out optic nerve atrophy. After a few days, the patient was discharged from the ICU to inpatient dermatology. His urinary tract infection was treated with antibiotics. After discharge from the hospital, we contacted the patient, who had returned to his normal state of health. Dapsone treatment was successfully replaced by oral Prednisolone at that time.

## 3. Discussion

MetHb-emia results from increasing transformation of the Hb-iron from a ferrous (Fe^2+^) into an oxidized ferric (Fe^3+^) state. Erythrocytes continuously produce low levels of MetHb secondary to autooxidation [3]. This disables the Hb molecule to bind oxygen and results in a left shift of the oxygen Hb dissociation curve [6]. However, MetHb usually comprises less than 1% of the total Hb in healthy individuals. Such low MetHb levels are achieved by the regulatory actions of nicotinamide adenine dinucleotide (NADH)–dependent cytochrome b5 reductase enzyme (about 67–95% of the conversion), nicotinamide adenine dinucleotide phosphate (NADPH)-dependent MetHb reductase (about 5% of the conversion) and the nonenzymatic antioxidants Ascorbate and Glutathione (about 12–15% of the conversion). Conversely, when the latter control processes are malfunctioning or overwhelmed, high MetHb concentrations in erythrocytes can be found [3].

There are certain congenital forms of MetHb-emia, whereas the acquired forms are usually observed in hospitals and are mostly induced by drugs. Indeed, previous reports have estimated that 4–13% of patients receiving Dapsone develop some degree of hemolytic anemia or MetHb-emia [3]. According to a recent US national poison data system study, intoxication with Dapsone is not uncommon: in 1209 cases of Methylene blue administration for clinically significant MetHb-emia, Dapsone was among the top four substance categories [7].

To the best of our knowledge, the herein presented clinical case of Dapsone-induced MetHb-emia is the first demonstrating severe symptoms (e.g., tachycardia, GCS 5, hypoxia-induced generalized seizure event, need for extensive invasive ventilation) at relatively low arterial MetHb concentrations with a peak at 6%. Remarkably, and in contrast to previous reports, this low-grade MetHb-emia induced significant symptoms in absence of any pre-existing major inflammatory, cardiac, pulmonary, or neurological conditions [8,9]. According to the majority of previous publications, oxygen saturation levels and severe clinical symptoms, as observed in our patient, do not typically become evident until MetHb concentrations of 20–25% are exceeded. For instance, headache, fatigue, tachycardia, weakness, and dizziness are normally observed at MetHb levels of 30–40%. When MetHb-emia approaches 60%, this can lead to arrhythmia, coma and seizures, while death is reported to occur at levels higher than 70% [3]. Even in the setting of pre-existing significant diseases, early signs of MetHb-emia usually are reported at MetHb levels of 10–15% [6,8,9,10,11,12]. Of note, it is known that the severity of MetHb-emia-related symptoms is relative to the patient’s Hb level [3]. We detected mild anemia in our patient, which has probably contributed to the observed scenario. Thus, although we observed atypically low arterial MetHb concentrations with excellent response to extensive mechanical ventilation, the coincidence of the newly initiated medication along with classic signs and symptoms, the lack of alternative explanations, and a rapid response to Methylene blue and Ascorbate treatment, strongly support the diagnosis of an aquired Dapsone-induced MetHb-emia. A recent review summarized that when receiving therapeutic doses of Dapsone, patients can become symptomatic at MetHb levels greater or less than 10% and that the existing MetHb level cut-offs for symptoms, mainly derived from overdose cases in otherwise young healthy individuals, are not representative of what is frequently seen in present-day clinical practice [3]. In general, the MetHb level is useful to confirm diagnosis, but it is not as important as the patient’s clinical status for determining early treatment [3].

In the liver, Dapsone undergoes reversible acetylation by N-acetyltransferase to monoacetyl Dapsone, or is alternatively metabolized in a cytochrome (CYP) P450 dependent manner by N-hydroxylation to Dapsone hydroxylamine. The hydroxylated compounds undergo glucuronidation and, in addition to the parent compound Dapsone, are excreted in urine. Importantly, the hydroxylamine metabolites are retained in the circulation for a long period, as they undergo enterohepatic recirculation [3]. According to the latter information, the half-life of Dapsone can vary between 14–83 h [4,5,9,10]. Due to the potentially prolonged half-life of Dapsone, complete recovery of the patient took several days. Moreover, although his liver function studies revealed mildly elevated enzyme levels (e.g., γ-glutamyl-transferase 1.88 (0.17–1.19) µmol/L*s), paraclinical indicators of hepatic synthetic and clearance functions were mainly within normal limits, except for cholinesterase, which was mildly compromised (83 (89–215) µmol/L*s). The latter was probably attributable to the presence of mild NAFLD in our patient [13,14]. Therefore, reduced protein binding, impaired metabolism and/or metabolite accumulation due to significant liver dysfunction is not a realistic scenario to explain the event as described in this case report.

The hydroxylamines resulting from Dapsone metabolism are quickly taken up by erythrocytes, where they are responsible for the formation of MetHb and hemolysis. It was shown that one hydroxylamine molecule can react with more than five molecules of Hb [3]. Our patient showed no evidence of G6PD-deficiency, and we consistently did not detect significant hemolysis. However, patients with normal G6PD-function are more susceptible to MetHb formation [4]. Methylene blue is reduced to Leukomethylene blue, which acts as an electron donor to enhance erythrocyte MetHb reduction [15]. In more detail, Methylene blue acts as a cofactor of the NADPH-dependent MetHb reductase by accepting an electron from NADPH, being reduced to leukomethylene blue. The latter donates the electron to MetHb and finally causes the conversion back to oxyhemoglobin [3]. However, Dapsone can induce rebound MetHb-emia within hours after Methylene blue treatment [16]. Therefore, we repeatedly administered Methylene blue along with Ascorbate as an adjunct. The latter is an unspecific, yet strong, chemical reducing agent. However, because of its relatively slow onset of action Ascorbate is not recommended as a monotherapy for acute symptomatic MetHb-emia [3]. Otherwise, Ascorbate has been shown to be a helpful adjunct in the treatment of MetHb-emia in previous clinical cases [4,12].

The patient was taking his medication on a regular basis, including Pantoprazole. Dapsone is predominantly metabolized by CYP2C9 and CYP2C19 [17,18]. According to the official German drug information guidelines, reduced clearance rates of Dapsone with Omeprazol as a regular co-medication are suggested, probably explained by interactions at the CYP2C19 isoenzyme level. CYP2C19 is considered to be the major enzyme in the metabolism of all proton pump inhibitors [19]. This explicitly includes Pantoprazole, although significant inter-individual variability due to different genotypes is acknowledged [20]. Therefore, we cannot exclude the possibility that co-medication of Pantoprazole and Dapsone could have at least contributed to mediating the clinical symptoms observed by the patient’s wife over the four-week treatment period. One potential scenario to explain the event as described above comprises reduced CYP2C19 metabolization of Dapsone due to Pantoprazole. This could have led to predominant hepatic transformation of Dapsone by CYP2C9 with consecutively rising hydroxylamine levels [17]. However, it ultimately remains unclear whether such a predominant metabolization of Dapsone by CYP2C9 with consecutively increased hydroxylamine formation and chronic low-grade MetHb-emia was indeed the main mechanism contributing to the herein reported clinical scenario. Otherwise, this hypothetical impact could help to explain the gradually increasing signs and symptoms reported by the patient’s wife.

## 4. Conclusions

Clinicians should be aware of the fact that Dapsone-induced MetHb formation can result in significant clinical symptoms over the course of a few short weeks, even at relatively low arterial concentrations of less than 10% and in absence of major pre-existing diseases. Co-medication with Pantoprazole or other proton pump inhibitors should be considered as a risk factor, which could accelerate MetHb formation.

## Figures and Tables

**Figure 1 biomedicines-10-02837-f001:**
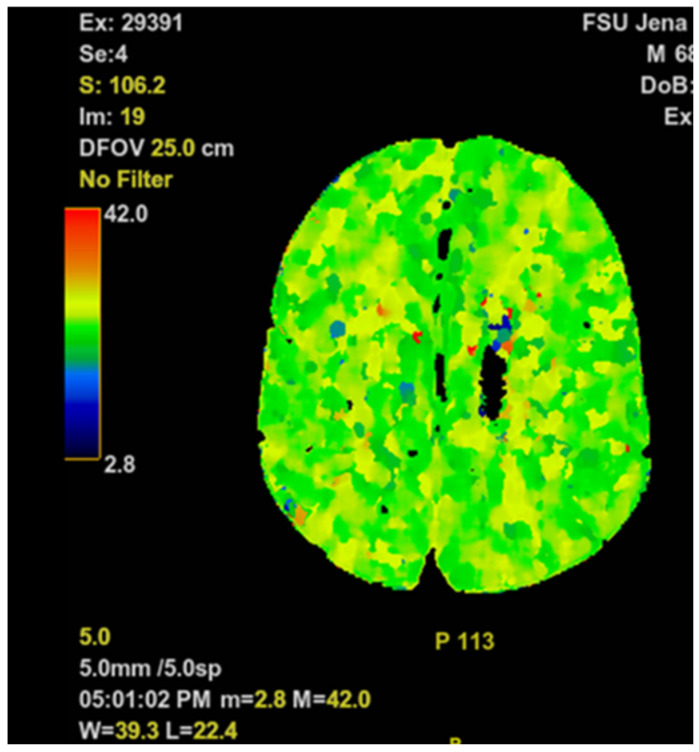
Representative cerebral perfusion scan slice of the head imaging study.

**Figure 2 biomedicines-10-02837-f002:**
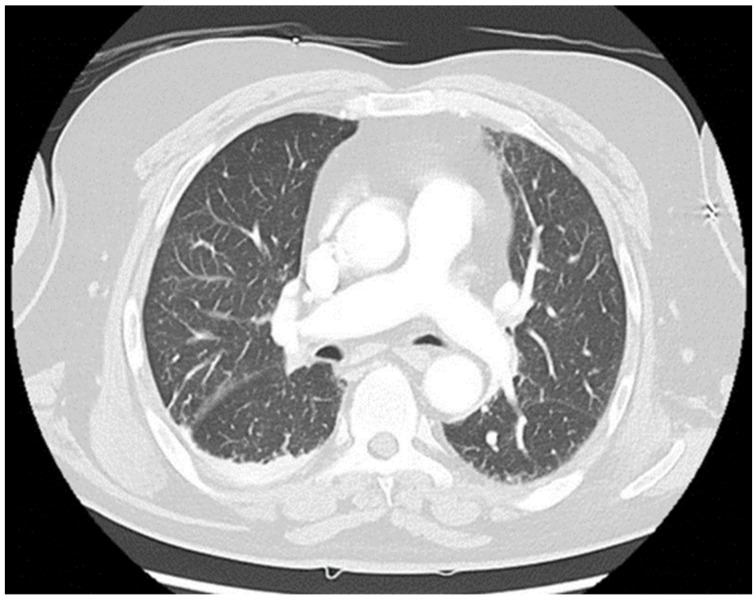
Representative CT scan slice of the chest imaging study.

**Table 1 biomedicines-10-02837-t001:** Representative arterial blood gas analyses of the patient in relation to invasive mechanical ventilation.

Parameter	Day 1			Day 2		Day 3	
paO2 [kPa]	58.9	39.6	30.6	10.9	16.7	12.7	14.8
paCO2 [kPa]	4.4	4.3	4.4	5.1	4.9	5.2	4.5
pH	7.44	7.45	7.45	7.43	7.43	7.42	7.45
Lactate [mmol/L]	1.7	0.9	0.8	1.0	0.7	0.8	0.8
Hb [mmol/L]	6.9	6.6	6.5	6.9	6.2	6.5	6.0
MetHb [%]	6.0	2.2	2.3	2.6	2.5	2.8	1.9
Ventilation Pmax [mB]	24	20	20	15	-	-	-
Ventilation PEEP [mB]	9	10	10	5	-	-	-
Ventilation FiO2 [%]	100	70	70	40	-	-	-

## Data Availability

Not applicable.

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
