# Peer review of "Hypoxia in A Patient with Anti-p200 Pemphigoid under Combined Dapsone and Pantoprazole Treatment"

_biomedicines, 2022, doi:10.3390/biomedicines10112837_

Round 1

Reviewer 1 Report

This work by Lang et al describes a case report of a dapsone adverse drug reaction. The manuscript is very well laid out, and very well written.

I have only 4 minor comments, regarding minor text edits, and other than that I'm recommending aceptance after minor revision.

Line 34 – “ Mainly hydroxylamines”, perhaps something along the lines of “Dapsone metabolites are mainly hydroxylamine” would sound better

Line 49 – Please clarifyif any treatment was administered immediately upon admission.

Lines 192-194 – This sentence is confusing. 2C19 does not produce hydroxylamines but 2C9 does? Perhaps a little rewording might help.

 Lines 195-197 – I’m not sure this sentence has a clear meaning. There is no doubt that hydroxylamines are able to form MetHb (for example, https://doi.org/10.1016/B978-0-12-386454-3.00856-3). Is this what the authors believe requires “future research” ?

Author Response

Line 34 – “ Mainly hydroxylamines”, perhaps something along the lines of “Dapsone metabolites are mainly hydroxylamine” would sound better

  • We have paraphrased the sentence according to the referee’s suggestion. The adaptions are now highlighted in yellow.

Line 49 – Please clarifyif any treatment was administered immediately upon admission.

  • No specific treatment was initiated, since the seizure event was observed quickly after admission to inpatient dermatology. We have added this information to the current manuscript.

Lines 192-194 – This sentence is confusing. 2C19 does not produce hydroxylamines but 2C9 does? Perhaps a little rewording might help.

  • Th referee is right. We have accordingly paraphrased this paragraph.

 Lines 195-197 – I’m not sure this sentence has a clear meaning. There is no doubt that hydroxylamines are able to form MetHb (for example, https://doi.org/10.1016/B978-0-12-386454-3.00856-3) . Is this what the authors believe requires “future research” ?

  • We are thankful for the referee’s hint, since we were not aware of the confusing nature of this sentence for the reader. There is indeed no doubt that hydroxylamines induce MetHb formation. However, whether a pharmacological interaction of Dapsone and Pantoprazole at the CYP-level were the main mechanisms contributing to explain the oberved clinical scenario in our patient, remains speculative. In our view future research will have to elucidate the significance of such potential interactions for MetHb formation. However, we have paraphrased the respective paragraph.   

Reviewer 2 Report

The manuscript is good clinical report and important observation. I have following doubts

1) Since Hb from day 1 till day 3 did not fluctuate even after decrease of paO2 concentration. This observation failed to discuss thoroughly in the discussion. 

2) Did clinician observed ferroptosis in the patient

3) 

Author Response

  • Since Hb from day 1 till day 3 did not fluctuate even after decrease of paO2 concentration. This observation failed to discuss thoroughly in the discussion.
  • We thank the referee for the important hint. We have now added information on mechanical ventilation to table 1, to enable the reader to better relate paO2 and FiO2 to fluctuations of Hb and Met-Hb over the treatment course. We, moreover, paraphrased the respective paragraph.

  • Did clinician observed ferroptosis in the patient
  • This is an interestig point. However, we did not observe ferroptosis in this case, since it is e.g. characterized morphologically by the presence of ultrastructural mitochondrial alterations (DOI: 10.1038/cdd.2015.158). It would have been necessary to perform experimental molecular analyses for evaluating this pathway. Our report is, however, of clinical nature. Otherwise we recognize the referee’s point and will regard it in future studies.
